# Relationship Between Serum Uromodulin as a Marker of Kidney Damage and Metabolic Status in Patients with Chronic Kidney Disease of Non-Diabetic Etiology

**DOI:** 10.3390/ijms252011159

**Published:** 2024-10-17

**Authors:** Radmila Žeravica, Branislava Ilinčić, Dragan Burić, Ana Jakovljević, Veljko Crnobrnja, Dalibor Ilić, Marija Vukmirović Papuga

**Affiliations:** 1Department of Pathophysiology and Laboratory Medicine, Faculty of Medicine, University of Novi Sad, 21000 Novi Sad, Serbia; radmila.zeravica@mf.uns.ac.rs (R.Ž.); branislava.ilincic@mf.uns.ac.rs (B.I.); dragan.buric@mf.uns.ac.rs (D.B.); ana.jakovljevic@mf.uns.ac.rs (A.J.); veljko.crnobrnja@mf.uns.ac.rs (V.C.); 2Center of Laboratory Diagnostic, University Clinical Center of Vojvodina, 21000 Novi Sad, Serbia; 3Center for Radiology, University Clinical Center of Vojvodina, 21000 Novi Sad, Serbia; dr.dalibor.ilic@gmail.com; 4Department of Nuclear Medicine, Faculty of Medicine, University of Novi Sad, 21000 Novi Sad, Serbia

**Keywords:** chronic kidney disease, obstructive nephropathy, serum uromodulin, metabolic syndrome

## Abstract

Chronic kidney disease (CKD) is often associated with dyslipidemia, marked by lipid abnormalities that can worsen kidney function and increase cardiovascular risk. A promising biomarker for evaluating kidney function and metabolic status in chronic kidney disease (CKD) is serum uromodulin (sUmod). This study sought to further investigate the relationship between sUmod levels and metabolic status in non-diabetic CKD patients. A sensitive ELISA method was used to determine sUmod levels in 90 adults with obstructive nephropathy and 30 healthy controls. Kidney function was assessed using the measured glomerular filtration rate (mGFR) through renal clearance of 99mTc-diethylenetriamine penta-acetic acid, along with cystatin C levels. Additionally, glycemic and lipid statuses were evaluated. sUmod concentrations showed a significant association with High-density lipoprotein (HDL) levels. Furthermore, CKD patients with lower sUmod levels had significantly lower Apolipoprotein A-I (Apo A-I) values compared to the control group. Significant predictors of lower sUmod concentrations identified in this study were higher glycemia (B = −15.939; *p* = 0.003) and lower HDL cholesterol levels (B = 20.588; *p* = 0.019). We conclude that, in addition to being significantly reduced in CKD patients, sUmod is a potential predictor of metabolic syndrome (MS) in this population. Lower sUmod concentrations, independent of mGFR, predict lower HDL cholesterol levels and higher glycemia values.

## 1. Introduction

Chronic kidney disease (CKD) has emerged as a major worldwide health challenge in the 21st century. Owing to advancements and standardization of diagnostic methods and protocols worldwide, it has become possible to monitor morbidity and mortality from CKD on a global level. According to published studies, the total number of patients with CKD worldwide, including all stages (I–V), exceeds 846,600,000 [1]. The significant number of CKD patients contributes strongly to the morbidity and mortality from non-communicable diseases [2]. Data on the prevalence of non-diabetic CKD are limited, but some studies suggest that the prevalence is approximately 10% (non-diabetic non-hypertensive patients) [3,4].

CKD is a complex condition frequently associated with dyslipidemia, which consists of several abnormalities in the lipid profile. Lipid profile includes various parameters, the primary ones being the levels of total cholesterol, HDL-C (high-density lipoprotein cholesterol), LDL-C (low-density lipoprotein cholesterol), and triglycerides [5]. Among individuals affected by CKD, dyslipidemia consists of low HDL-cholesterol and elevated triglycerides level [6].

These abnormalities can exacerbate kidney function deterioration and contribute to the onset of cardiovascular complications [6]. Thus, assessing lipid profile in patients with CKD is of importance for their clinical monitoring and prognosis and adjustment of therapy.

One of the potential biomarkers in assessment of kidney function in CKD is uromodulin (Umod), a glycoprotein produced in the tubular cells of the kidney’s thick ascending limb of Henle’s loop [7,8]. Most of the produced Umod is secreted in the urine, where in healthy individuals, it predominates over other proteins. Uromodulin plays roles in anti-inflammatory protection and prevention of kidney stone formation, and it possess immunomodulatory function [7,8,9,10,11]. A remaining portion of Umod is released into the blood (serum uromodulin—sUmod) [12,13,14]. Since sUmod is thought to reflect tubular integrity and, indirectly, nephron mass and renal reserve, it emerges as a valuable marker of kidney function [13,14,15].

Furthermore, available data suggest that sUmod may serve as a biomarker for assessing metabolic status and the risk of metabolic complications among individuals suffering from CKD [16]. Reduced levels of Umod in CKD are also correlated with increased inflammation and oxidative stress, which further lead to lipid imbalance and lipoprotein dysfunction [17]. By influencing lipid profiles, sUmod becomes a potential biomarker that reflects the state of kidney function and the risk of total cardiometabolic complications.

This study aimed to further clarify the relationship between sUmod levels and metabolic status in individuals suffering from non-diabetic CKD.

## 2. Results

### 2.1. Baseline Characteristics

Table 1 presents the baseline characteristics of the study population. The entire study population was composed of white (Caucasian) participants. The control group and patients with non-diabetic CKD did not differ significantly in terms of mean age (t = 1.9209; *p* = 0.0672), nor did the groups examined differ significantly in terms of gender distribution (chi-square = 2.534; *p* = 0.114). 

### 2.2. Biochemical Evaluation

Study population was evaluated for inflammatory markers, electrolyte status and liver enzymes (Table 2).

Measured levels of inflammatory markers in patients with non-diabetic CKD differed significantly when compared to control group. Lower average concentrations of CRP (C-reactive protein) were detected in the control group (1.6 ± 0.9 µg/mL) compared to patients with CKD (2.5 ± 1.8 µg/mL) (*p* < 0.001). Additionally, significantly lower concentrations of fibrinogen were measured in the control group (3.0 ± 0.6 g/L) compared to the values in the CKD group (3.4 ± 0.8 g/L) (*p* < 0.05).

At the same time, there were no significant differences detected in electrolyte status (Ca, Ca^2+^, P and Mg) and liver enzymes (AST, ALT, GGT) between the control group and patients with CKD. All values in both groups were within reference ranges.

### 2.3. Renal Function Parameters

When the study population was evaluated for kidney function, as expected, significant differences were detected between the control group and patients with CKD (Table 3).

### 2.4. Metabolic Syndrome Parameters

When compared for glycemic status parameters, a significant difference was detected between the control group and patients with non-diabetic CKD. Significantly lower levels of blood glucose were detected in the control group (4.8 ± 0.5 mmol/L) compared to the CKD group (5.2 ± 0.6 mmol/L). A similar pattern was detected in HbA1C (%) levels (5.3 ± 0.5 in the control group vs. 5.6 ± 0.4 in the CKD patients) (Table 4).

Significant differences in lipid profile between the control and CKD groups were detected as well. Higher values of both HDL cholesterol and Apo A-I levels were detected in the control group compared to the CKD group (HDL 1.5 ± 0.4 mmol/L in CG vs. 1.3 ± 0.4 mmol/L in the CKD group, *p* = 0.0193; Apo A-I 1.6 ± 0.3 g/L in the CG vs. 1.4 ± 0.4 g/L in the CKD group (*p* = 0.0134)) (Table 4).

### 2.5. Uromodulin Analysis

The average value of sUmod concentration in study population is shown in Table 5. Significantly higher values were detected in control group compared to CKD patients (F = 20.660; *p* < 0.001).

There was no significant difference in sUmod level for sex as a single independent variable (F = 2.431; *p* = 0.122). Additionally, the interaction effect between sex and mGFR (CKD status) on sUmod level was not significant (F = 0.365; *p* = 0.547) (Table 6; Figure 1).

A positive correlation was detected between sUmod and mGFR (r = 0.83; *p* < 0.001) (Figure 2).

Multivariate analysis (sUmod as dependent variable) is shown in Table 7. Variables that were significant at the 0.05 level in the univariate models were included in the multivariate linear regression model. Significant predictors of higher sUmod concentrations were higher HDL cholesterol level (B = 20.588; *p* = 0.019) and lower blood glucose level (B = −15.939; *p* = 0.003).

## 3. Discussion

In our previous study, uromodulin came to light as potential noninvasive biomarker of early kidney disfunction in individuals suffering obstructive nephropathy [18]. Other recent studies suggest that higher levels of sUmod can be related to favorable metabolic profile, apart from other positive impacts on homeostatic regulation [15].

The aim of the conducted research was to explore the correlation between sUmod levels and metabolic status in patients with non-diabetic CKD. The key finding is that sUmod concentrations are significantly associated with HDL levels. Additionally, patients with CKD and lower sUmod levels exhibited significantly reduced Apo A-I values compared to the control group, while differences in other lipid status parameters were not significant. The significance of these findings stems from the fact that the lower levels of both lipid parameters (HDL and Apo A-I) in the CKD group suggest a diminished atheroprotective capacity, which, together with the lower uromodulin levels, places these patients at a higher cardiovascular risk.

In CKD, it is assumed that uromodulin has a protective effect, considering the fact that in CKD patients the production of uromodulin per nephron unit is increased [19,20]. In patients with kidney disease in its early stages, when there is no visible decline in kidney function, Umod synthesis is increased [21,22]. However, with the progression of fibrosis and the loss of functional nephrons, uromodulin production decreases, and consequently, uromodulin levels drop as GFR declines and CKD advances [13,19,23]. In the current study, significantly lower Umod levels were detected in patients with CKD compared to the healthy population and showed a significant correlation with GFR values. This is consistent with previous research, which has also demonstrated the significance of uromodulin as a predictor of reduced functional kidney mass. The average serum concentration of Umod in patients with non-diabetic CKD was 48.5 ± 27.2 ng/mL and was significantly lower compared to the healthy subjects, which was 80.6 ± 21.5 ng/mL (t = 4.938; *p* < 0.001). Similar differences in sUmod concentrations were presented in the study by Fedak et al., with measured concentrations higher than those in our study—healthy subjects 191.2 ng/mL (89.1–299.1 ng/mL), CKD patients 68.8 ng/mL (38.2–109.9 ng/mL) (*p* < 0.001) [24].

In terms of differences between sUmod levels in female and male individuals, there was no significant difference in the sUmod levels between men and women in CKD and healthy individuals both, although higher levels were detected in female participants in our study. These results are in line with findings of recent study by Nanamatsu et al., which suggest that there are higher sUmod levels in female participants, based on estrogen responsiveness of Umod gene in examined model [25]. Lack of significant differences in our study population between sUmod level in male and female participants might be explained by the fact that the most of female participants were women older than child-bearing age, when levels of estrogen naturally decline.

Given that MS is linked to an elevated risk of cardiovascular and kidney complications, and that CKD in MS can progress to end-stage kidney disease, our goal was to look at the possibilities of using uromodulin as a predictor of metabolic status in patients with CKD, independent of GFR.

In this study, higher BMI values were detected in the non-diabetic CKD group (26.3 kg/m^2^) compared to the control group of healthy subjects (23.9 kg/m^2^). This finding supports the thesis that obesity and CKD are positively correlated, as evidenced by the decline in GFR [26,27]. The presence of excessive adipose tissue, especially in cases of central obesity, is associated with a decrease in GFR and kidney function impairment [28]. It is important to note, though, that metabolic conditions like diabetes, dyslipidemia, hypertension, and inflammation are frequently present in conjunction with obesity, which makes it challenging to separate the role of obesity in the onset and progression of CKD [26,27].

Regarding lipid profile, significantly higher values of HDL cholesterol were detected in control group (1.5 ± 0.4 mmol/L) compared to CKD group (1.3 ± 0.4 mmol/L), with Apo A-I levels also being higher in the healthy subjects. It is known that low HDL cholesterol levels have been linked to a higher risk of GFR decline [29,30]. As HDL cholesterol levels decrease, HDL cholesterol’s anti-inflammatory and anti-apoptotic properties are lost [31,32,33]. In patients with CKD, which is characterized, among other things, by chronic inflammation, HDL levels are not only reduced, but HDL is also structurally altered and characterized by compromised anti-inflammatory activity [34].

Significantly higher values of CRP and fibrinogen concentrations in our study support this thesis (CRP = 2.5 ± 1.8 µg/mL; fibrinogen 3.4 ± 0.8 g/L in CKD group vs. CRP = 1.6 ± 0.9 µg/mL; fibrinogen 3.0 ± 0.6 g/L in control group). These findings are in line with the widely accepted theory that a variety of pro-inflammatory mediators are essential to the intricate pathophysiology of chronic kidney disease (CKD), impacting the illness’s start and course via a complex network of interactions [35,36,37]. Moreover, lower levels of uromodulin in patients with CKD may further reduce protective anti-inflammatory extent, considering, among other mechanisms, its importance in the activation of interleukin 23/17 [38].

When assessing glycemic profile in our study population, the values of examined parameters were within the reference range in both the control and CKD groups. However, lower blood glucose and HbA1C levels were detected in healthy individuals (4.8 ± 0.5 mmol/L; 5.3 ± 0.5%) compared to the CKD group (5.2 ± 0.6 mmol/L; 5.6 ± 0.4%). These results align with findings in other studies. Diabetes is recognized to have an impact on the onset and progression of CKD and has been extensively studied [35,39,40,41]. Nevertheless, even in normoglycemic individuals like the subjects in our study, higher blood glucose concentrations, though still within the reference range, are associated with a higher risk of CKD onset and progression [23,42,43].

A significant correlation between sUmod concentrations and some of the MS parameters was found in patients with non-diabetic CKD. In a multivariate linear regression model higher glycemia (B = −15.939; *p* = 0.003) and lower HDL cholesterol values (B = 20.588; *p* = 0.019) were identified as significant predictors of lower sUmod concentrations.

Apart from the effect of glycemia on renal function, dyslipidemias [44] and atherosclerosis [45] are also linked to the decline in kidney function. Our model did not identify any of the following as significant predictors of sUmod concentrations: LDL (*p* = 0.150), triglycerides (*p* = 0.360), Apo A-I (*p* = 0.061), and Apo B (*p* = 0.216). However, HDL cholesterol level was identified as a predictor; lower concentrations are associated with lower sUmod concentrations and, subsequently, lower GFR values.

In the KORA F4 study, prevalent MS was inversely associated with sUmod concentration. However, the study failed to prove a relationship between sUmod and the development of MS or any of its components (incident MS). The authors propose that additional research is necessary to clarify the interplay between sUmod levels and MS, specifically whether the drop in sUmod levels (which suggests tubular injury) is a sign of a reduced renal reserve brought on by a decrease in nephron mass or if some other aspect of MS is affecting uromodulin biosynthesis [46].

The interplay of kidney disease, sUmod level and MS was recently also reported in a study by Scheurlen et al., in which metabolic surgery leads to increased sUmod levels, suggesting improvement in ‘kidney health’ or renal reserve, which traditional kidney function biomarkers could not detect [47].

An advantage of sUmod as a biomarker in early nephropathy was reported by Leiherer et al., who revealed that sUmod levels had an inverse association with type 2 diabetes, independent of estimated GFR. Therefore, the authors suggest that sUmod may serve as a tissue-specific marker for the early stages of diabetic nephropathy, which is not captured by estimates of GFR due to glomerular hyperfiltration associated with diabetes [48].

Some recent studies also suggest that positive Umod correlation with HDL cholesterol, besides negative correlation, might indicate the potential of Umod to modulate lipid metabolism and thus reduce the coronary heart disease risk [49,50]. Similar results were presented by Delgado et al., in which higher sUmod concentrations were linked to more favorable metabolic profiles, a decreased prevalence of comorbidities, and a reduced mortality risk over a 10-year period [16].

Based on the results of our study and in relevance to above mentioned studies of other authors we conclude that, apart from being considerably decreased in individuals suffering from CKD, sUmod is also a potential predictor of MS in the same population. Reduced sUmod concentrations, independent of GFR, are predictors of lower HDL cholesterol levels and higher glycemia values.

### Limitations of the Study

The conducted research has several limitations. Firstly, the limitation regarding the recruitment of patients from only one center, with a sample size that was not exceptionally large, restricts the ability to generalize the obtained results. Additionally, there was an incompleteness of certain MS parameters.

There are emerging studies pointing importance of uromodulin in the pathogenesis of salt sensitive hypertension [51]. Although arterial hypertension is a significant component of the metabolic syndrome, we had limited access to data regarding parameters as blood pressure values, the duration of hypertension and concomitant antihypertensive therapy, hence this is one of the limitations of the study.

The listed limitations would require robust testing in clinical practice to ultimately determine the clinical relevance of the study results.

The advantage of the conducted study compared to other similar studies is that we used 99 mTc-labeled diethylenetriamine penta-acetic acid (99 mTc-DTPA) clearance to determine GFR as an indicator of kidney function, while other studies typically use one of the available formulas to calculate GFR based on the concentrations of available biomarkers. Another advantage is the use of sUmod, which has been shown to be a superior biomarker to urinary uromodulin in other studies.

## 4. Materials and Methods

The cross-sectional study, carried out from 2019 to 2022 at the Clinical Center of Vojvodina (CCV), was approved by the CCV Ethics Committee in compliance with the Declaration of Helsinki. Informed consent, provided in writing, was secured from each participant.

### 4.1. Study Subjects

Ninety patients were included in the study population; they were referred for renal scans and assessments of radioisotope clearance to evaluate renal function, both split and total, in relation to obstructive nephropathy (ON). Following the K/DOQI criteria, each participant was diagnosed with CKD after a comprehensive evaluation that included clinical examinations, biochemical analysis, and imaging abnormalities [52].

Individuals with an mGFR < 15 mL/min/1.73 m^2^, a solitary kidney, diabetes mellitus, chronic liver disease, acute infectious or inflammatory illnesses, or cancer were not included in the study.

The control group comprised 30 healthy individuals, potential kidney donors. Renal scans and clearance tests were used to evaluate kidney function.

### 4.2. Study Protocol

All participants obtained a physical examination which included body height and weight measurements, later used to calculate the BMI. Physical examination was followed by urine and fasting venous blood sampling and kidney function assessment.

### 4.3. Measurement of Uromodulin Concentration

Blood samples for sUmod quantification collected during the study visit were stored at −80 °C until analysis. Serum uromodulin concentrations were quantified using a human Umod ELISA kit (CLOUD-CLONE CORP., Katy, TX, USA), in compliance with the guidelines provided by the manufacturer.

### 4.4. Assessment of Glycemic and Lipid Profile

Using an Abbott Architect c8000 biochemical analyzer (Abbott Laboratories, Abbott Park, IL, USA), the standard enzyme-specific GOD-PAP technique was used to determine glycemia.

The determination of HbA1C was performed using a two-step immunological test for the quantitative determination of HbA1C concentration in whole blood samples. It was carried out using the chemiluminescent microparticle immunoassay (CMIA) with flexible test protocols (Chemiflex) on the automated ARCHITECT ci4100 analyzer (Abbott Laboratories, Abbott Park, IL, USA) utilizing commercial kits from the same manufacturer.

Lipid profiles were determined using a standard enzymatic procedure on the automated biochemical analyzer ARCHITECT ci4100 from Abbott Laboratories, with commercial kits from the same manufacturer. Serum concentrations of apolipoproteins A-1 and B were determined immunoturbidimetrically on the OLYMPUS apparatus (Olympus, Tokyo, Japan), using commercial tests from the same manufacturer.

### 4.5. Biochemical and Electrolyte Analysis

Levels of AST, ALT, and gamma GT were determined using a kinetic UV test on the automated biochemical analyzer Abbott Architect c8000, utilizing commercial kits from the same manufacturer.

Determination of urea, creatinine, and uric acid levels was performed using a kinetic UV test on the automated biochemical analyzer Abbott Architect c8000, with commercial kits from the same manufacturer.

Determination of fibrinogen concentration was performed using the Clauss coagulation method on an ACL analyzer with commercial kits from Instrumentation Laboratory, Bedford, MA, USA.

High-sensitivity C-reactive protein (hsCRP) levels were measured using a latex-enhanced immunoturbidimetric technique on Siemens Healthcare Diagnostics ADVIA^®^ 1800 Chemistry (Siemens Healthcare Diagnostics, Tarrytown, NY, USA) automated biochemical analyser, with commercial kits from the same manufacturer.

Determination of electrolytes was performed using flame photometry on an ILyte Instrumentation Laboratory flame photometer with commercial kits from the same manufacturer.

### 4.6. Kidney Function Assessment

The following parameters were measured to precisely assess kidney function:(a)GFR—The method by Fleming et al. [53] was used to determine GFR through isotopic clearance of 99 mTc-DTPA. Paper chromatography was used to assess the radiochemical purity of the isotopes.(b)Biomarkers of renal function

Cystatin C concentrations and serum and urine creatinine levels, along with serum urea and uric acid concentrations, were measured using the methods previously reported by Vukmirovic Papuga et al. [18].

### 4.7. Statistical Analysis

The data description was displayed as n (%), mean ± standard deviation. Pearson’s and Spearman’s correlation coefficients were used to estimate the correlation between variables. A two-way between-groups analysis of variance was conducted to explore the impact of sex and mGFR (CKD status) on uromodulin level. To model the relationship of dependent variables with potential predictors, a linear regression model was used. Significant predictors (significance level of 0.05) from univariate analyses were later included in the multivariate regression models. IBM SPSS Statistics 26.0 (SPSS Inc., Chicago, IL, USA) software package was used to perform all statistical analyses.

## Figures and Tables

**Figure 1 ijms-25-11159-f001:**
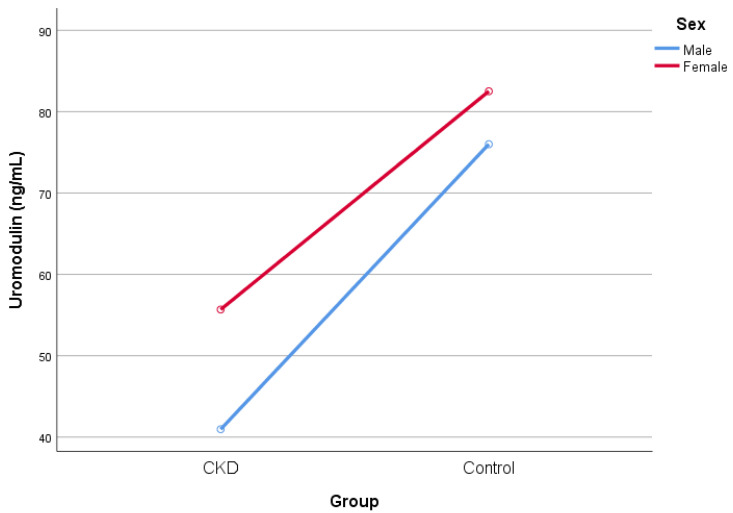
A two-way between-groups analysis of variance to explore the impact of sex and mGFR (measured glomerular filtration rate) (CKD status) on sUmod (serum uromodulin) level.

**Figure 2 ijms-25-11159-f002:**
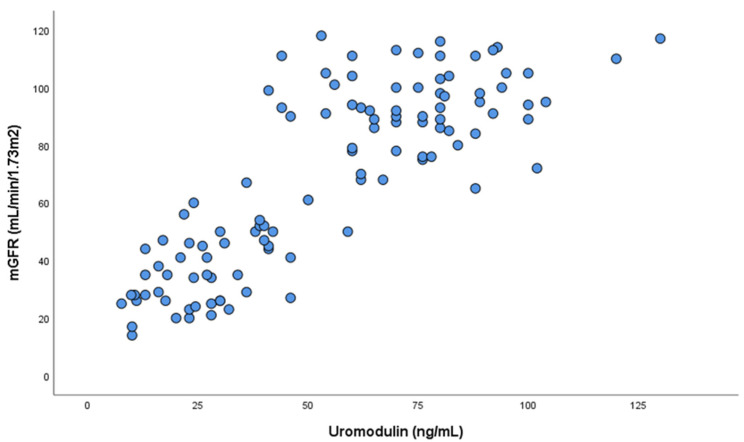
Correlation between sUmod (serum uromodulin) concentration and mGFR (measured glomerular filtration rate).

**Table 1 ijms-25-11159-t001:** Baseline characteristics of study population.

	CG (N = 30) $\bar{x}$ ± SD	CKD (N = 90) $\bar{x}$ ± SD	*p*
Age	50.7 ± 14.30	56.1 ± 12.87	0.0672
Female/male (%)	9/21	46/44	0.0712

CG—control group, CKD—chronic kidney disease.

**Table 2 ijms-25-11159-t002:** Inflammatory markers, electrolyte status, and liver enzymes in study population.

	CG (N = 30) $\bar{x}$ ± SD	CKD (N = 90) $\bar{x}$ ± SD	*p*
CRP (µg/mL)	1.6 ± 0.9	2.5 ± 1.8	<0.001
Fibrinogen (g/L)	3.0 ± 0.6	3.4 ± 0.8	0.0134
Ca (mmol/L)	2.4 ± 0.1	2.4 ± 0.1	>0.05
Ca^2+^ (mmol/L)	1.2 ± 0.04	1.2 ± 0.1	>0.05
P (mmol/L)	1.1 ± 0.2	1.1 ± 0.2	>0.05
Mg (mmol/L)	0.8 ± 0.04	0.8 ± 0.1	>0.05
ALT (U/I)	23.5 ± 10.9	22.5 ± 6.8	>0.05
AST (U/I)	21.8 ± 6.8	23.2 ± 5.9	>0.05
GGT (U/I)	22.5 ± 8.3	22.0 ± 8.0	>0.05

CG—control group, CKD—chronic kidney disease.

**Table 3 ijms-25-11159-t003:** Renal function parameters in study population.

	CG (N = 30) $\bar{x}$ ± SD	CKD (N = 90) $\bar{x}$ ± SD	*p*
mGFR (mL/min/1.73 m^2^)	107.6 ± 7.7	60.1 ± 27.9	<0.001
S Creatinine (µmol/L)	70.3 ± 7.9	116.8 ± 64.8	<0.001
Urea (mmol/L)	4.6 ± 1.4	7.6 ± 4.6	<0.001
Uric Acid (µmol/L)	274.6 ± 51.2	370.3 ± 100.6	<0.001
Cystatin C (mg/L)	0.76 ± 0.07	1.4 ± 0.7	<0.001

CG—control group, CKD—chronic kidney disease, mGFR—measured glomerular filtration rate.

**Table 4 ijms-25-11159-t004:** Metabolic syndrome parameters in study population.

	CG (N = 30) $\bar{x}$ ± SD	CKD (N = 90) $\bar{x}$ ± SD	*p*
BMI (kg/m^2^)	23.9 ± 1.3	26.3 ± 1.4	0.0583
Blood glucose level (mmol/L)	4.8 ± 0.5	5.2 ± 0.6	0.0013
HbA1C (%)	5.3 ± 0.5	5.6 ± 0.4	0.0011
Cholesterol (total) (mmol/L)	5.1 ± 1.0	5.1 ± 1.0	>0.05
HDL cholesterol (mmol/L)	1.5 ± 0.4	1.3 ± 0.4	0.0193
LDL cholesterol (mmol/L)	2.8 ± 0.9	3.0 ± 0.8	>0.05
Triglycerides (mmol/L)	1.3 ± 0.9	1.6 ± 1.0	>0.05
Apo A-I (g/L)	1.6 ± 0.3	1.4 ± 0.4	0.0134
Apo B (g/L)	0.9 ± 0.3	1.0 ± 0.3	>0.05

CG—control group, CKD—chronic kidney disease, BMI—body mass Index, HbA1C—glycated hemoglobin, Apo A-I—apolipoprotein A-I, Apo B—apolipoprotein B.

**Table 5 ijms-25-11159-t005:** sUmod concentration in study population.

	CG (N = 30) $\bar{x}$ ± SD	CKD (N = 90) $\bar{x}$ ± SD	*p*
Uromodulin (ng/mL)	80.6 ± 21.5	48.5 ± 27.2	*p* < 0.001

CG—control group, CKD—chronic kidney disease.

**Table 6 ijms-25-11159-t006:** sUmod concentration in male and female study participants.

	Sex	CG (N = 30) $\bar{x}$ ± SD	CKD (N = 90) $\bar{x}$ ± SD	*p*
Uromodulin (ng/mL)	Male	76.00 ± 16.02	40.96 ± 26.10	
	Female	82.50 ± 23.80	55.68 ± 26.47	0.547

**Table 7 ijms-25-11159-t007:** Multivariate analysis (sUmod—dependent variable).

	Univariate		Multivariate	
	B	*p*	B	*p*
Age (years)	−0.647	0.001	−0.310	0.223
Sex (female/male)	16.772	0.062		
Cholesterol (total)	6.396	0.035	1.959	0.540
HDL cholesterol	26.324	<0.001	20.588	0.019
LDL cholesterol	5.508	0.150		
Triglycerides	−2.937	0.360		
Apo A-I	19.495	0.061		
Apo B	14.851	0.216		
CRP	−4.135	0.107		
Fibrinogen	−5.365	0.119		
Blood glucose level	−21.718	<0.001	−15.939	0.003
HbA1C	−16.784	0.098		
BMI	−0.786	0.188		

Apo A-I—apolipoprotein A-I, Apo B—apolipoprotein B, CRP—C-reactive protein, HbA1C—glycated hemoglobin, BMI—body mass index.

## Data Availability

The data presented in this study are available upon reasonable request from the corresponding author.

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
