# Peer review of "Relationship Between Serum Uromodulin as a Marker of Kidney Damage and Metabolic Status in Patients with Chronic Kidney Disease of Non-Diabetic Etiology"

_ijms, 2024, doi:10.3390/ijms252011159_

Round 1
Reviewer 1 Report
Comments and Suggestions for Authors
The study demonstrates that sUmod levels are significantly reduced in individuals with CKD and could potentially serve as a valuable predictor of MS in the population studied.
The figure legends could benefit from more detailed descriptions.
The reference list should be updated to include more recent studies, as only one paper from 2024 was cited.
I also strongly recommend presenting a more detailed demographic table, showing variables such as sex, age, race, and weight, along with blood pressure data, if available.
Given the increasing emphasis on sex and gender differences in scientific research, incorporating this aspect would enhance the paper’s relevance. Including sex/gender data can help identify disparities, improve understanding of biological differences, and provide valuable insights that may lead to more effective strategies in healthcare and education. (Clayton JA, Tannenbaum C. Reporting sex, gender, or both in clinical research? Jama. 2016 Nov 8;316(18):1863-4.)
So, the authors should discuss and show the data splitting the sexes and state any differences observed in the results.
Reviewer 2 Report
Comments and Suggestions for Authors
ijms-3210056
Relationship Between Serum Uromodulin as a Marker of Kidney Damage and Metabolic Status in Patients with Chronic Kidney Disease of Non-Diabetic Etiology.
This is an interesting, properly conducted study that aimed to further explore the correlation between serum uromodulin levels and metabolic status in non-diabetic chronic kidney disease patients, in 90 adults with obstructive nephropathy.
However, the relevance of these results in patients´ clinical practice remains to be determined. Furthermore, some other comorbidities other than or additional to chronic kidney disease have not been taken into account.
It might be interesting to increase the discussion with his previous article from 2022, highlighting the new approaches achieved with this manuscript, in order to reveal the great originality and novelty of this manuscript.
In addition, many interesting References in this field, corresponding to the years 2023 and 2024, are missing.
Round 2
Reviewer 2 Report
Comments and Suggestions for Authors
ijms-3210056 Cr2
Relationship Between Serum Uromodulin as a Marker of Kidney Damage and Metabolic Status in Patients with Chronic Kidney Disease of Non-Diabetic Etiology.
Once the authors have already addressed every point, I have no more concerns.